SOFTWARE

# OmnibusX: A unified platform for accessible multi-omics analysis

**Linh Truong, Thao Truong, Huy Nguyen**◉*

OmnibusXLab, OmnibusX Company Limited, Ho Chi Minh City, Vietnam

* huy@omnibusx.com

## Abstract

OmnibusX is an integrated, privacy-centric platform that enables code-free multi-omics data analysis by bridging computational methodologies with user-friendly interfaces. Designed to overcome challenges posed by fragmented analytical tools and high computational barriers, OmnibusX consolidates workflows for diverse technologies - including bulk RNA-seq, single-cell RNA-seq, single-cell ATAC-seq, and spatial transcriptomics - into a single, cohesive application. The application integrates established open-source tools such as Scanpy, DESeq2, SciPy, and scikit-learn into transparent, reproducible pipelines, offering users control over analytical parameters. Additionally, OmnibusX features proprietary modules, including a highly accurate cell-type prediction engine and an interactive plotting editor for generating publication-quality visualizations. Available as a standalone desktop application and an enterprise edition for centralized server deployment, OmnibusX ensures all data processing is conducted locally, eliminating external data transfer and usage tracking. By lowering technical barriers and enhancing reproducibility, OmnibusX aims to accelerate biological discovery and foster robust, data-driven collaborations. A fully documented trial version is accessible at: https://omnibusx.com/apps.

## Author summary

We developed OmnibusX to address the analytical challenges posed by modern high-dimensional biological datasets, including single-cell RNA sequencing (scRNAseq), single-cell ATAC sequencing (scATACseq), bulk RNA sequencing (bulk RNAseq), and spatial transcriptomics. These technologies provide complementary insights into gene expression, chromatin accessibility, and spatial organization, but their analysis often requires multiple software tools, advanced programming skills, and complex workflows, limiting accessibility and reproducibility.

OmnibusX unifies these analyses within a single, privacy-focused desktop platform, integrating established computational methods with proprietary

**Data availability statement:** All relevant data are within the manuscript and its Supporting Information files.

**Funding:** The author(s) received no specific funding for this work.

**Competing interests:** The authors have declared that no competing interests exist.

modules through an intuitive graphical interface. By consolidating modality-specific pipelines, interactive visualization, and customizable publication-ready outputs, the platform enables researchers to perform comprehensive analyses without coding. This approach reduces technical barriers, promotes reproducibility, and facilitates collaborative, data-driven biological discovery.

## Introduction

Recent advances in multi-omics technologies have significantly expanded the scope of biological research by enabling detailed investigation of cellular states, molecular mechanisms, and tissue-level organization. Techniques such as bulk RNA sequencing, single-cell RNA sequencing (scRNA-seq), single-cell ATAC sequencing (scATAC-seq), and spatial transcriptomics each provide complementary views of gene expression, chromatin accessibility, and spatial context across biological systems..

However, analyzing these data types remains computationally challenging. Researchers often rely on disparate software tools, command-line interfaces, and complex scripting workflows, which can pose significant barriers for researchers without programming expertise. These fragmented pipelines also raise challenges for reproducibility. Moreover, concerns related to data privacy and institutional policies, especially in clinical or regulatory environments, further restrict the use of cloud-based solutions, highlighting the need for secure, locally executable alternatives.

To address these challenges, we developed OmnibusX, a unified and privacy-centric platform for multi-omics data analysis. OmnibusX integrates widely adopted open-source packages with proprietary analytical modules into a graphical interface that does not require programming. By supporting end-to-end workflows for multiple omics technologies within a single system, OmnibusX eliminates common barriers in multi-omics analysis, enhances reproducibility, and empowers a broader range of researchers to conduct robust, data-driven investigations.

## Design and Implementation

### Architecture and design

OmnibusX is a cross-platform desktop application comprising two primary components: a local analytics server and a graphical user interface (GUI) client. The analytics server, developed in Python, handles data management, preprocessing, and execution of analytical functions. The GUI client, built using Electron (v29.1.0) and React (v18.2.0), delivers a consistent and interactive experience across operating systems. OmnibusX supports native execution on Windows, macOS (both Intel and Apple Silicon), and Ubuntu Linux. Platform-specific builds are compiled and distributed separately to ensure consistent performance across environments. A complete list of software packages and dependencies is provided in S1 Table in S1 Data.

The internal architecture follows established software engineering principles, emphasizing modularity, fault tolerance, and platform isolation. Input data is

organized into a structured local file system by the Python server for efficient retrieval and reuse. When a user initiates an analysis via the GUI, only the relevant data subset is processed, and the results are returned for visualization through interactive plots and dashboards. This division between backend computation and frontend rendering ensures responsive performance while minimizing memory and compute overhead.

All analytical processing is performed locally, with communication between components occurring via a local API. To maintain strict data privacy, no input data or analytical results are transferred externally. Communication with the central OmnibusX server, hosted on Amazon Web Services (AWS), is limited to user authentication, license validation, and downloading necessary reference files (e.g., gene annotations and curated marker sets).

OmnibusX integrates core analytical functions from widely used open-source frameworks such as Scanpy [1] and Seurat [2], ensuring performance consistent with these established tools. Benchmarking results for Scanpy's or Seurat's underlying methods are available in its original publication, providing detailed reference points for computational efficiency and scalability.

Job execution is managed through a custom multiprocessing task manager built on Python's Queue interface. This manager schedules compute-intensive steps, monitors resource usage (memory and CPU), and automatically terminates jobs exceeding safe limits. Users can view and cancel tasks through the application interface. On the desktop version, analyses must run with the application open. In the enterprise edition, tasks are executed on a private centralized server, allowing users to disconnect and resume later.

To accommodate users with limited local computational resources while preserving data privacy, OmnibusX offers enterprise deployment on a private centralized server under the user's control. This server can be hosted either on-premises within the institution's infrastructure or on the user's dedicated cloud environment (e.g., AWS, Google Cloud, or equivalent). In all cases, computations are executed entirely on hardware owned or managed by the user, and all data remain strictly within their administrative domain. Users connect to the centralized server through the OmnibusX client, enabling secure, high-capacity computation without transferring data outside the user-controlled environment. This approach ensures scalability while meeting institutional and organizational data-governance requirements.

Comprehensive technical documentation—including detailed descriptions of all integrated open-source packages, their role in preprocessing and downstream analysis, and the interaction between proprietary and open-source components—is publicly available at: https://omnibusx.com/resources#user-guide.

## Gene annotation

Gene identifiers are standardized using Ensembl release version 111 (https://ftp.ensembl.org/pub/release-111/). For human datasets, gene symbols and IDs from older genome assemblies (GRCh37/hg19) are automatically mapped to their corresponding entries in GRCh38/hg38. When multiple aliases or outdated identifiers refer to the same gene, raw counts are aggregated before normalization to eliminate redundancy and ensure accurate interpretation. Annotation files are downloaded automatically from the OmnibusX server based on the user-specified species and stored locally for future use, minimizing redundant downloads.

## Quality control

Quality control (QC) is performed on user-submitted datasets immediately upon upload. Metrics such as total counts, number of detected features, and mitochondrial read percentage are computed and visualized. These visualizations are interactive, allowing users to set QC thresholds based on real-time feedback from the distribution of these metrics. The raw, unfiltered dataset is preserved in storage to allow reprocessing under different thresholds without requiring re-upload.

## Modality-specific processing pipelines

OmnibusX provides structured, modality-specific processing pipelines built on the Scanpy framework. These pipelines are designed to process raw data through consistent and reproducible analytical steps, tailored to the characteristics of

scRNA-seq, scATAC-seq, spatial transcriptomics (including 10X Visium, Visium HD, NanoString GeoMx, and Slide-seq), and bulk RNA-seq datasets.

Default normalization strategies are selected automatically based on the input data type and can be adjusted as needed. For scRNA-seq, scATAC-seq, Slide-seq, and 10X Visium HD, log normalization is applied. Antibody-derived tag (ADT) data are normalized using centered log-ratio (CLR) transformation. For bulk RNA-seq and NanoString GeoMx datasets, trimmed mean of M-values (TMM) normalization is used for dimensional reduction analysis, while transcripts per million (TPM) normalization is applied for visualization.

After normalization, highly variable gene selection is performed to focus downstream analyses on biologically informative features. Principal component analysis is then applied to the normalized expression matrix, and non-linear dimensionality reduction is performed using UMAP or t-SNE on the top 50 principal components. For bulk RNA-seq and NanoString GeoMx data, PCA is computed on all genes, and the first two principal components are used for exploratory scatter plots. Clustering is performed using the Leiden algorithm, with a default resolution of 0.8. This parameter can be adjusted through the user interface.

For scRNA-seq, scATAC-seq multiome, and Visium HD datasets from human or mouse, OmnibusX automatically applies its cell type prediction model to provide initial cell type annotations. These predictions are based on OmnibusX's curated markers and serve as a foundation for further manual refinement or subgroup analysis.

If the input includes hashtag oligo (HTO) data, OmnibusX automatically performs HTO demultiplexing using a Python implementation of the Seurat HTODemux algorithm (https://satijalab.org/seurat/reference/htodemux). Demultiplexing results are stored as cell metadata.

For scATAC-seq data, chromatin accessibility regions are parsed from feature names formatted as chromosome: start-end. Minor variations in genomic coordinates for the same peak across samples can result in inconsistent feature naming, introducing batch effects and redundancy. To address this, OmnibusX automatically merges overlapping regions across samples and re-annotates them as the longest contiguous interval encompassing all overlaps. Redundant intervals are consolidated into single feature entries. DNA fragment coordinates are then indexed using interval trees, enabling fast retrieval of all fragments within any queried genomic region. Chromosome-level gene annotations for the relevant species are downloaded automatically and used to display genomic context alongside accessibility peaks in OmnibusX's integrated genome browser.

For immune profiling, OmnibusX supports the import of TCR/BCR contig annotation files in Cell Ranger format. These are parsed, indexed, and used for clonotype identification, classification, and downstream analyses, including clonotype diversity, repertoire overlap, and spectratyping within defined cell groups.

For spatial transcriptomics data, OmnibusX supports both H&E-stained images and IF images. Images are indexed using a multi-resolution pyramid structure to enable high-performance rendering, interactive zooming, and region-specific queries. For fluorescence images, OmnibusX supports real-time channel merging and intensity adjustment. Additionally, OmnibusX supports the alignment of individual sample images into a unified spatial grid, allowing for intuitive navigation and cross-sample comparison within a consistent coordinate framework.

## Cell annotation and selection

OmnibusX includes a built-in cell type prediction engine for scRNA-seq, scATAC-seq, and Visium HD datasets from human and mouse. This module is based on a curated database of 166 cell type and subtype-specific marker gene sets, derived from 280 publications and validated across multiple datasets and tissues. The prediction algorithm builds on the AUCell method with three main steps: (1) removing low-level background signals, (2) calculating enrichment scores for each marker set and assigning cell labels based on the highest enrichment, and (3) smoothing unassigned cells by nearest-neighbor consensus. This process enables accurate identification of closely related subtypes and robust performance across diverse datasets. The methodology, algorithms, and benchmarking results are described in detail in Truong

et al., 2024 [3]. While the curated marker sets and source code are proprietary, users can adjust the resulting cell type annotations manually or apply their own custom marker sets as the knowledge base for prediction, using the same procedure as the proprietary module.

Users may select cells flexibly through three mechanisms: filtering based on gene expression thresholds, filtering by metadata combinations (e.g., cell type, condition), or manual selection using a lasso tool on UMAP/t-SNE embeddings. Additionally, multi-level subclustering is supported, allowing focused analysis of specific cell subsets through iterative refinement.

Metadata management tools allow users to create, modify, or delete metadata fields associated with cells or samples. This enables structured comparisons and data stratification. Users can also upload and manage custom gene sets for use in enrichment analysis or module scoring. To support consistent figure aesthetics, OmnibusX provides a color palette manager that allows researchers to define and reuse customized palettes across datasets and analyses.

## Batch effect correction

OmnibusX supports batch effect correction using both the Harmony algorithm (via the HarmonyPy Python implementation) and the ComBat method available in Scanpy. Corrections are applied to the principal component space and are used exclusively for visualization purposes (t-SNE, UMAP, and clustering) to improve interpretability across batches. By reducing variation caused by technical or experimental differences, batch correction aligns biologically similar cells from different batches more closely in the embedding space, making cell populations more directly comparable across samples. Importantly, all statistical testing and downstream analyses are performed on the original raw or normalized expression values to preserve biological integrity and avoid introducing artifacts from over-correction.

## Differential expression analysis

OmnibusX supports flexible differential expression analysis for both single-cell and bulk RNA-seq datasets. For single-cell data, users can perform statistical comparisons using either t-tests or the Wilcoxon rank-sum test. For pseudobulk or bulk datasets, DESeq2 is executed via the PyDESeq2 Python wrapper.

To facilitate large-scale comparisons, OmnibusX includes automated workflows for differential expression, such as pairwise testing across all cell types or condition-specific comparisons within each cell population. Results are displayed through interactive dashboards that combine statistical tables with visualizations, including volcano plots, MA plots, violin plots, scatter plots, and summary statistics.

Differentially expressed genes (DEGs) are used directly within OmnibusX for other analyses, such as enrichment analysis and heatmap generation. For scATAC-seq datasets, differential chromatin accessibility can be visualized by linking DEG results to chromosome regions, providing integrative insights into gene regulation.

## Enrichment analysis

OmnibusX includes three methods for gene set enrichment analysis: GSEA (via GSEApy), Fisher's exact test, and AUCell, which is implemented in Python based on the original R algorithm. Users may select gene sets from curated databases such as MSigDB or input custom gene sets.

Results can be visualized through a variety of plots, including enrichment dot plots, z-score heatmaps, enrichment scatter plots, and mean expression plots. These outputs provide multiple perspectives for interpreting pathway-level activity across conditions or clusters.

## Heatmap analysis

OmnibusX provides a flexible and interactive heatmap module for expression-based comparisons across genes, cell populations, and experimental conditions. Heatmaps can be generated through multiple workflows: (i) Automatic marker

selection, performed for each cell group versus all other groups using differential expression analysis via t-tests or Wilcoxon rank-sum tests. Genes are ranked by statistical significance and fold change, with top-ranked markers selected for visualization. All statistical parameters are configurable through the user interface. (ii) Manual gene set input, allowing users to define and visualize custom gene lists across selected groups or metadata-defined categories. (iii) Pairwise sample clustering, based on expression profiles from bulk RNA-seq or pseudobulk-aggregated single-cell data. In this mode, OmnibusX computes pairwise distances between samples and generates a heatmap with an accompanying hierarchical clustering dendrogram. Supported distance metrics include Euclidean, Manhattan, Maximum, Canberra, and Binary, implemented via the SciPy [4] library. All heatmaps are interactive and customizable through a built-in configuration panel.

## Trajectory analysis

Trajectory and pseudotime inference in OmnibusX is enabled through native integration with the Palantir Python package. This module models differentiation dynamics within single-cell datasets by computing pseudotime, entropy, and cell fate probabilities, allowing users to reconstruct lineage relationships and study temporal gene expression changes.

The results are organized into multiple interactive dashboards, each designed to support a distinct aspect of trajectory exploration. The trajectory visualization dashboard displays the inferred differentiation paths overlaid on low-dimensional embeddings (e.g., UMAP or t-SNE), with coloring options for pseudotime, entropy, gene expression, or any user-defined metadata. The fate probability dashboard shows the likelihood of individual cells committing to specific terminal states, facilitating the study of lineage bias and divergence. Additional views include cell-state density plots, trajectory-aligned gene expression heatmaps showing dynamic transcriptional changes along pseudotime, and feature dynamics plots comparing gene behavior across different branches. These tools enable researchers to interrogate both global and branch-specific trends, supporting the identification of key regulators and lineage-defining events.

## TCR/BCR analysis

OmnibusX supports comprehensive T-cell and B-cell receptor (TCR/BCR) repertoire analysis for datasets containing immune receptor sequencing information. Upon importing annotated contig files (generated by Cell Ranger), OmnibusX extracts clonotype information and labels each cell with metadata including the number of detected receptor pairs, receptor type (TCR or BCR), detected chains (e.g., TRA, TRB, TRD, IGH, IGL, IGK), and clone size based on clonotype IDs. These attributes are integrated into the cell label, allowing users to immediately visualize and stratify cells based on clonotype characteristics. In-depth repertoire analysis is organized across several interactive dashboards: (i) Diversity analysis: Calculates clonotype diversity metrics using Shannon entropy, enabling users to assess immune repertoire complexity across groups or conditions. (ii) Repertoire overlap: Quantifies shared clonotypes between groups using Jaccard similarity of CDR3 sequences, supporting comparisons between cell groups. (iii) Spectratyping: Visualizes CDR3 length distributions and displays consensus CDR3 sequences within selected cell groups. These analyses provide insight into clonal expansion, immune activation, and lineage diversity, making them helpful in immuno-oncology, infectious disease, and vaccine response studies.

## Cell–cell interaction analysis

OmnibusX includes a built-in module for inferring cell–cell interactions in spatial transcriptomics datasets, leveraging curated ligand–receptor (L–R) pairs from CellPhoneDB. Users begin by specifying a cell group label (e.g., cell type or spatial cluster) to define the interacting regions. Interactions are identified based on the co-expression of ligand and receptor genes across these groups, and interaction strength is computed using the combined expression levels of both components within each interacting pair. To prioritize biologically relevant interactions, OmnibusX sorts the detected ligand–receptor pairs using ANOVA tests, evaluating differences in interaction strength across user-defined groups. Detected

interactions are further annotated with Gene Ontology (GO) terms to associate them with broader biological processes. Results are presented through interactive dashboards that display the spatial expression patterns of selected ligand–receptor pairs across tissue sections, supporting comparative analysis across conditions or regions.

### Plot customization tools

OmnibusX offers a highly configurable visualization environment that supports both automated and user-defined figure generation. For each analysis performed, results are rendered as context-specific plots (e.g., violin plots, bar plots, heatmaps, UMAP overlays), with corresponding configuration settings automatically generated and loaded into the plot editing module. This allows users to further refine the appearance and structure of visual outputs with minimal effort.

In addition to analysis-linked plots, users can independently generate custom plots using any combination of available gene expression data and metadata from the study. Each plot type, such as violin, bar, box, dot, composition, or scatter, has a dedicated configuration panel that supports flexible customization of how data is aggregated (e.g., sum, count, average, median) and displayed. Users can add auxiliary elements such as error bars, data points, category labels, and adjust element types (e.g., stacked vs. grouped bars).

All visual elements within a plot are independently stylable through a dedicated styling panel. Customization options include color palette selection, font style and size, axis and label formatting, layout settings, line thickness, marker size, element layering, and plot dimensions. These controls allow users to refine figures according to publication standards or presentation preferences directly within the application.

### Integrated genome browser

OmnibusX includes a built-in genome browser that enables interactive exploration of chromatin accessibility data in a genomic context. Designed as a lightweight alternative to the UCSC Genome Browser [5], it allows users to visualize accessibility peaks alongside annotated gene structures and compare patterns across cell groups or experimental conditions.

Peaks can be grouped and displayed by cell group label, enabling comparative analysis of chromatin accessibility across clusters, conditions, or sample groups. Two visualization modes are available depending on the size of the queried genomic region. For regions larger than 1 megabase (Mb), accessibility data are rendered as a heatmap, highlighting broad chromatin activity patterns and regional hotspots. For regions smaller than 1 Mb, the browser presents a detailed view showing gene structures, including introns, exons, and transcriptional direction, alongside chromatin accessibility peaks, displayed as histograms. Each cell group is shown as a separate track, facilitating direct visual comparison of accessibility profiles.

This module allows researchers to link differential chromatin accessibility to genomic features, investigate regulatory elements, and contextualize ATAC-seq signals within gene architecture, all within the OmnibusX environment.

## Results

### Architecture and design

OmnibusX is a modular, cross-platform desktop application consisting of a local analytics server (Python) and a graphical user interface (using Electron and React), supporting native execution on Windows, macOS, and Ubuntu Linux. The analytics server organizes and processes input data locally, while the client provides an interactive interface for triggering analyses and visualizing results through plots and dashboards. Communication between components occurs via a local API, ensuring responsive performance and preserving data privacy. Core analyses are executed entirely on the user's machine; remote interaction with the central OmnibusX server (hosted on AWS) is limited to user authentication, license validation, and downloading external reference files such as gene annotations or marker sets. A schematic of the system architecture is shown in Fig 1. These interactions are performed only when required, are not persistent connections, and do not record any user logs, ensuring that scientific data and usage patterns remain entirely local.

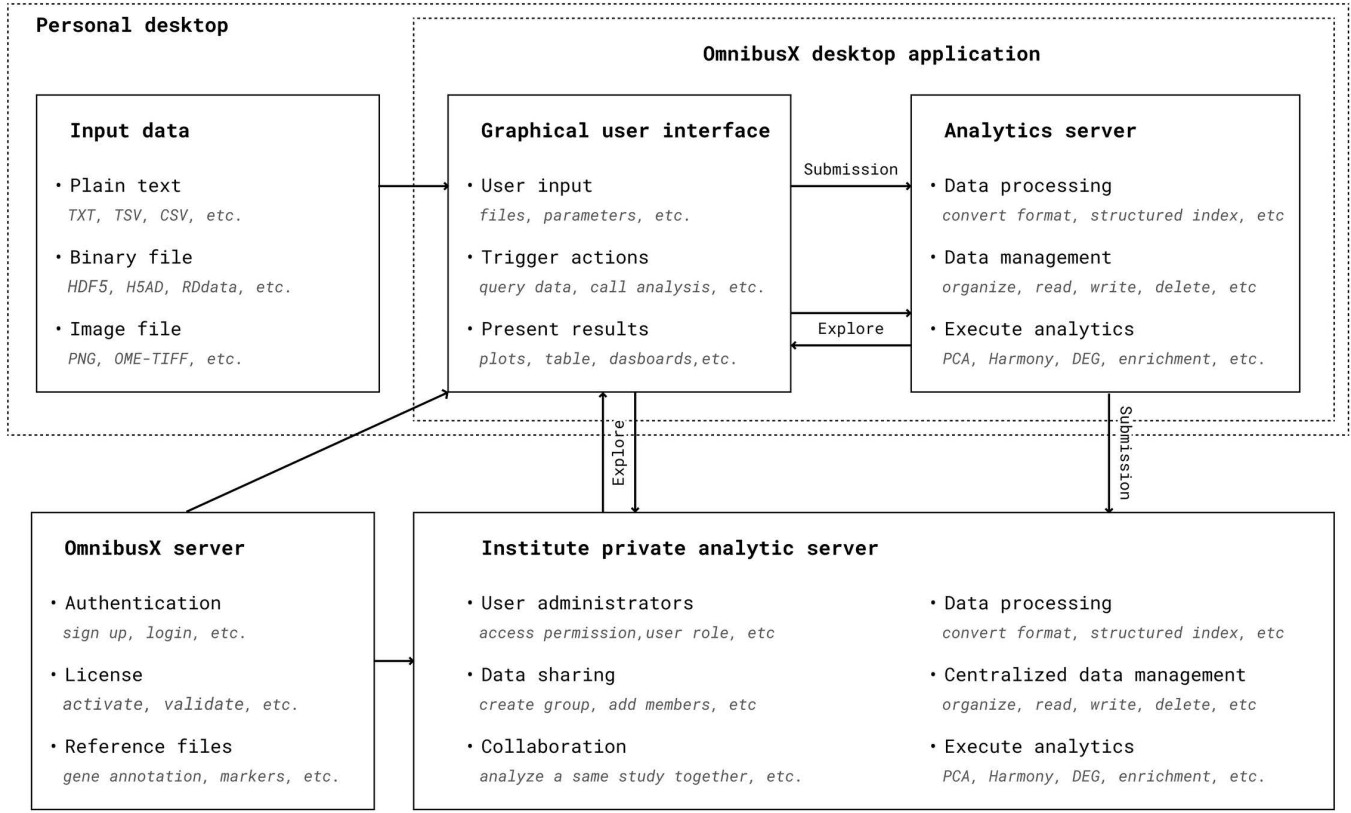

**Fig 1. Schematic architecture of OmnibusX, illustrating its modular design and data flow.** Users begin by submitting input data through the graphical user interface (GUI), which initiates backend processing on the Python analytics server. The server converts input files into structured formats, organizes them, and stores them locally on the user's machine. OmnibusX manages these files internally, allowing users to trigger specific analysis functions via the GUI. Only the necessary subset of data is loaded and processed on demand, with results returned to the GUI and rendered as interactive tables, plots, and dashboards. OmnibusX can also connect to a private analytics server within an institution, enabling centralized data sharing and collaborative analysis. In both standalone and enterprise configurations, communication with the central OmnibusX server is limited to user authentication, license validation, and downloading reference files such as gene annotations or marker sets.

## Input data and format conversion

Different omics quantification methods generate different data types, each with specific storage formats for the underlying molecular modality. These range from gene expression matrices to chromatin accessibility fragments, histological (H&E) or multiplexed immunofluorescence (IF) images, assembled TCR/BCR sequences, etc. Even when encoding similar biological information, different platforms and quantification tools produce heterogeneous file formats, including plain text (CSV/TSV), image files (PNG, JPEG, TIFF, OME-TIFF), and binary formats (HDF5, H5AD, RDS, etc.). Furthermore, the internal structure of binary files can vary significantly depending on the originating tool or pipeline, for example, between a 10X Genomics Cell Ranger HDF5 file and a Scanpy [1] HDF5 file, and different versions of Seurat [2] objects.

Handling these diverse formats typically requires researchers to learn the specifications of each file type and develop custom scripts for data extraction and conversion, presenting a substantial bottleneck in preprocessing. OmnibusX addresses this challenge by providing built-in support for a wide range of commonly used data formats. Supported input formats in the current OmnibusX release (version 1.0.49) include:

PLOS Computational Biology

- scRNA-seq: Matrix Market files (mtx + tsv/csv) or HDF5 formats from 10X Genomics Cell Ranger; Scanpy (.h5ad); Seurat (.rds, v3–v5); CSV/TSV count matrices.

- scATAC-seq: 10X Genomics fragment files (.tsv.gz); peak count matrices (mtx + tsv/csv); Scanpy (.h5ad); Seurat (.rds, v3–v5).

- Spatial transcriptomics (10X Visium & Visium HD, NanoString GeoMX): Gene expression matrices (HDF5 or mtx + tsv/csv); spatial coordinates files; image files (PNG, JPEG, TIFF, OME-TIFF); Scanpy (.h5ad); Seurat (.rds, v3–v5).

- Bulk RNA-seq: Count matrices (CSV/TSV).

- TCR/BCR data: 10X Genomics Cell Ranger VDJ contig annotation files (.csv).

Upon data upload, OmnibusX automatically identifies, extracts, and converts only the necessary components required for downstream analysis, while preserving the original data in a secure, read-only format.

### Gene name annotation

Expression count generated by common alignment tools often report gene-level counts using gene identifiers (e.g., Ensembl IDs), while downstream analyses and biological interpretation typically require gene symbols. Additionally, when integrating samples from different sources, datasets may contain a mixture of gene IDs and gene names, resulting in inconsistent annotations and potential ambiguity..

OmnibusX addresses this issue by implementing an automated gene name standardization procedure. All gene identifiers are mapped to Ensembl gene annotation [6] as the reference. In cases where multiple identifiers map to the same gene, raw counts are aggregated before the normalization step.

### Quality control step

Quality control (QC) is an essential step for data cleaning by excluding outliers that may bias downstream results. Thresholds for QC parameters, such as minimum gene counts or percentage of mitochondrial genes, can vary significantly across datasets. This process typically involves multiple rounds of refinement until the remaining data meet the required quality standards. OmnibusX addresses this challenge by providing an interactive QC interface that visualizes the distribution of key metrics, enabling users to explore their data and adjust filtering parameters with real-time feedback. Metrics such as the number of detected genes, total counts per cell or sample, and mitochondrial content are presented through dynamic plots. As users modify threshold values, the system immediately reflects the impact on dataset size and composition, facilitating transparent and informed decision-making in the QC process.

### Modality-specific data processing pipelines

High-dimensional omics data analysis generally follows a series of steps, including normalization, selection of highly variable features, dimensionality reduction (e.g., PCA), embedding visualization (e.g., t-SNE or UMAP), and clustering. While this framework is shared across many data types, each omic modality requires distinct methodological choices and parameter configurations at different stages. For example, log normalization is commonly applied to single-cell RNA-seq data, whereas centered log-ratio (CLR) normalization is preferred for antibody-derived tag (ADT) counts. In bulk RNA-seq workflows, Trimmed Mean of M-values (TMM) normalization is often employed for dimensionality reduction analysis, while transcripts per million (TPM) values are more suitable for visualization purposes. OmnibusX addresses these differences by providing carefully optimized, modality-specific default pipelines for each supported technology, while also allowing users full flexibility to adjust parameters as needed.

Beyond general processing, OmnibusX includes specialized modules designed for handling modality-specific data structures. For scRNA-seq, scATAC-seq multiome, and Visium HD data, OmnibusX integrates a proprietary Cell type prediction [3] module, offering automated and accurate cell annotation for both human and mouse datasets. For spatial data,

OmnibusX supports the alignment of individual sample images into a unified coordinate grid, facilitating intuitive navigation and cross-sample comparison. All spatial images are indexed using a multi-resolution pyramid structure, allowing efficient retrieval of image tiles and interactive zooming, useful features for pathological analysis. For scATAC-seq data, OmnibusX indexes DNA fragment coordinates as interval trees, enabling visualization of chromatin accessibility peaks with OmnibusX's genome browser. Additionally, the platform supports TCR/BCR repertoire analysis from the contig annotation files, generated by Cell Ranger, allowing downstream clonotype-level investigation.

Finally, OmnibusX supports interoperability with externally processed data. Users can directly import processed datasets in Scanpy (.h5ad) or Seurat (.rds) formats, bypassing the built-in pipelines while taking full advantage of OmnibusX's downstream analysis and visualization capabilities. This flexibility ensures smooth integration into diverse laboratory and computational workflows.

## Downstream analyses

Following initial processing, biological interpretation requires iterative interaction with the data through a range of analytical functions. This exploration phase is critical for extracting meaningful insights, evaluating hypotheses, and understanding unexpected observations. Ideally, biologists, who possess direct knowledge of experimental design and biological context, should be able to independently navigate this phase. However, the need for coding expertise often presents a significant barrier. Relying on bioinformaticians for implementation introduces delays due to back-and-forth communication, limiting the pace of discovery.

To address this, OmnibusX provides a comprehensive suite of interactive, code-free tools that allow researchers to engage directly with their data. Exploration typically begins with data annotation, especially cell type labeling and tissue region annotation, which underpins all downstream comparative analyses and interpretations. OmnibusX offers multiple approaches for annotation: a proprietary cell type prediction module based on OmnibusX's curated marker sets; the ability to import and manage users' custom gene sets for automated labeling; and flexible manual annotation tools. As illustrated in Fig 2, users can visualize gene expression from multiple modalities (e.g., RNA, ATAC, ADT) on shared embeddings (UMAP or t-SNE), perform clustering, identify marker genes, and label cells using expression thresholds, metadata filters, or freehand lasso selection. Subsets of interest can be extracted for focused analysis. For spatial transcriptomics, OmnibusX facilitates direct interaction with high-resolution tissue images, integrating histological context with gene expression overlays to support region-specific annotation based on both morphological and molecular features. The spatial viewer is optimized for large-scale image handling, featuring multi-resolution pyramid indexing, real-time zoom and pan, channel overlay for fluorescence images, intensity adjustment, and interactive region selection via a built-in lasso tool. These capabilities enable precise, spatially informed annotation and enhance pathology-guided molecular interpretation.

Once annotations are established, OmnibusX provides a broad suite of tools for comparative analysis (Fig 3). These include compositional analysis across experimental groups and gene expression visualization using violin plots, bar charts, box plots, and bubble plots. Users can identify marker genes specific to cell clusters or experimental conditions directly within the embedding space (e.g., UMAP or t-SNE) or spatial context. Differential expression analysis can be performed between two conditions using statistical methods appropriate to the data type, such as t-tests and Wilcoxon rank-sum tests for single-cell data, and DESeq2 [7] for bulk or pseudobulk data. To streamline large-scale comparisons, OmnibusX supports automated differential expression workflows, including pairwise comparisons across all cell types or condition-specific comparisons within each cell population. Enrichment analyses can be conducted on the resulting DEGs using gene set enrichment analysis (GSEA [8]), Fisher's exact test, or AUCell [9], leveraging MSigDB [10] databases. These analyses can be applied to two-group comparisons or extended to multiple groups using ANOVA-based approaches. Heatmaps are supported through multiple workflows: automatic generation from top-ranked markers, hierarchical clustering of bulk or pseudobulk expression matrices, or manual construction based on selected gene sets and cell groupings. For trajectory inference, OmnibusX integrates Palantir [11], enabling intuitive exploration of differentiation

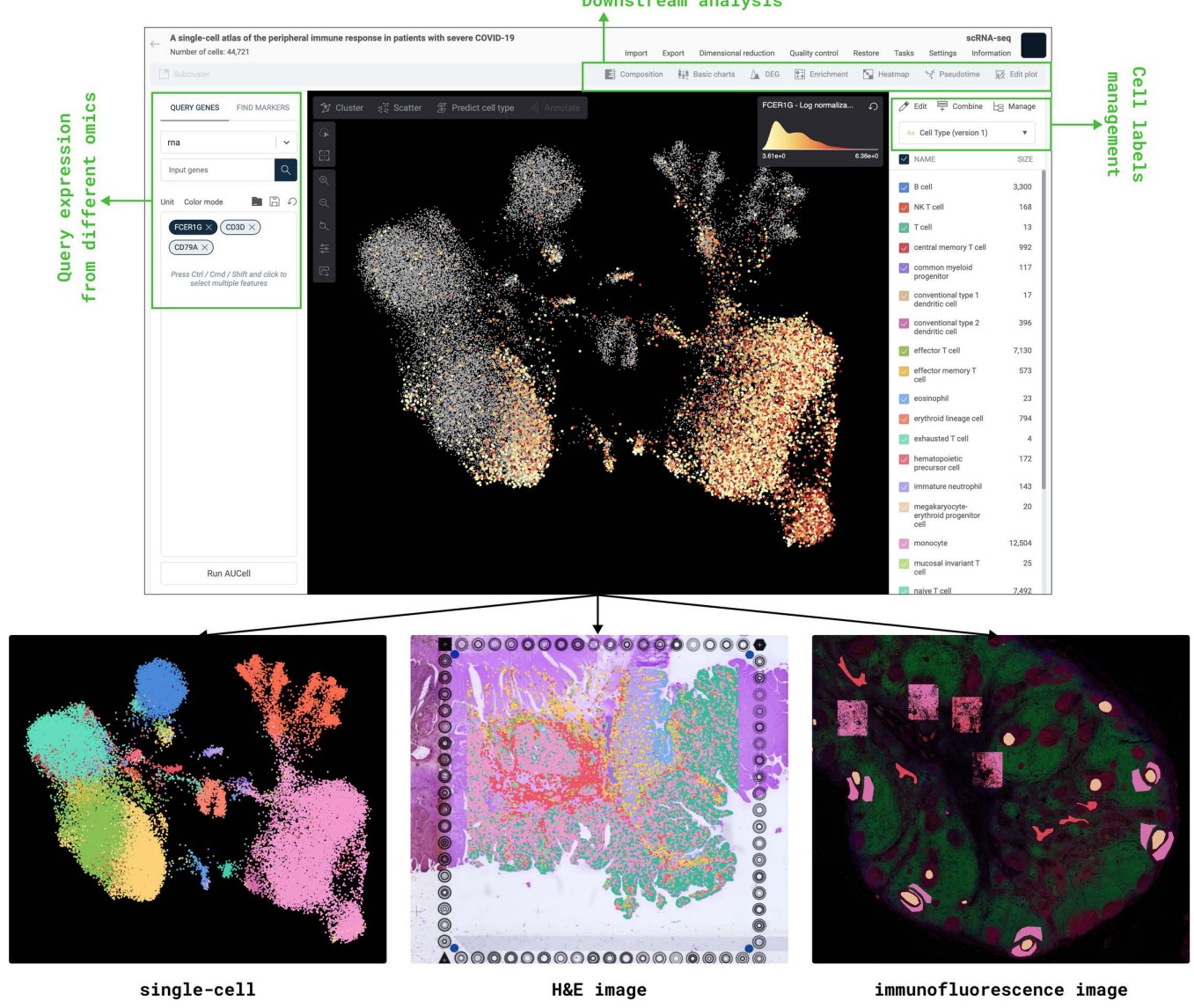

**Fig 2. Main interface of OmnibusX for dataset exploration.** The platform supports synchronized visualization of diverse data types, including cell embeddings (e.g., UMAP/t-SNE), H&E-stained tissue sections, and multiplexed immunofluorescence (IF) images. Users can interactively query gene expression across multiple omics modalities (e.g., RNA, ATAC, ADT), visualize results on embeddings, and manage or assign cell labels using built-in annotation tools. The interface is a central hub for intuitive data exploration, offering immediate access to downstream analytical modules, including compositional analysis, differential expression, enrichment, heatmap, trajectory inference, genome browser, clonotype analysis, etc.

pathways and lineage relationships. Researchers can visualize branching structures and infer pseudotemporal dynamics associated with development or disease.

For datasets composed of multiple samples, batch correction can be performed using Harmony. Corrected groupings are reflected in visualization embeddings (e.g., UMAP or t-SNE), while downstream analyses remain based on the original raw or normalized values to preserve analytical integrity. OmnibusX also provides technology-specific analysis tools. For scATAC-seq data, DNA fragment coordinates are indexed and visualized using a built-in genome browser (Fig 4), allowing inspection of chromatin accessibility peaks in the context of genomic annotations. For immune profiling, the

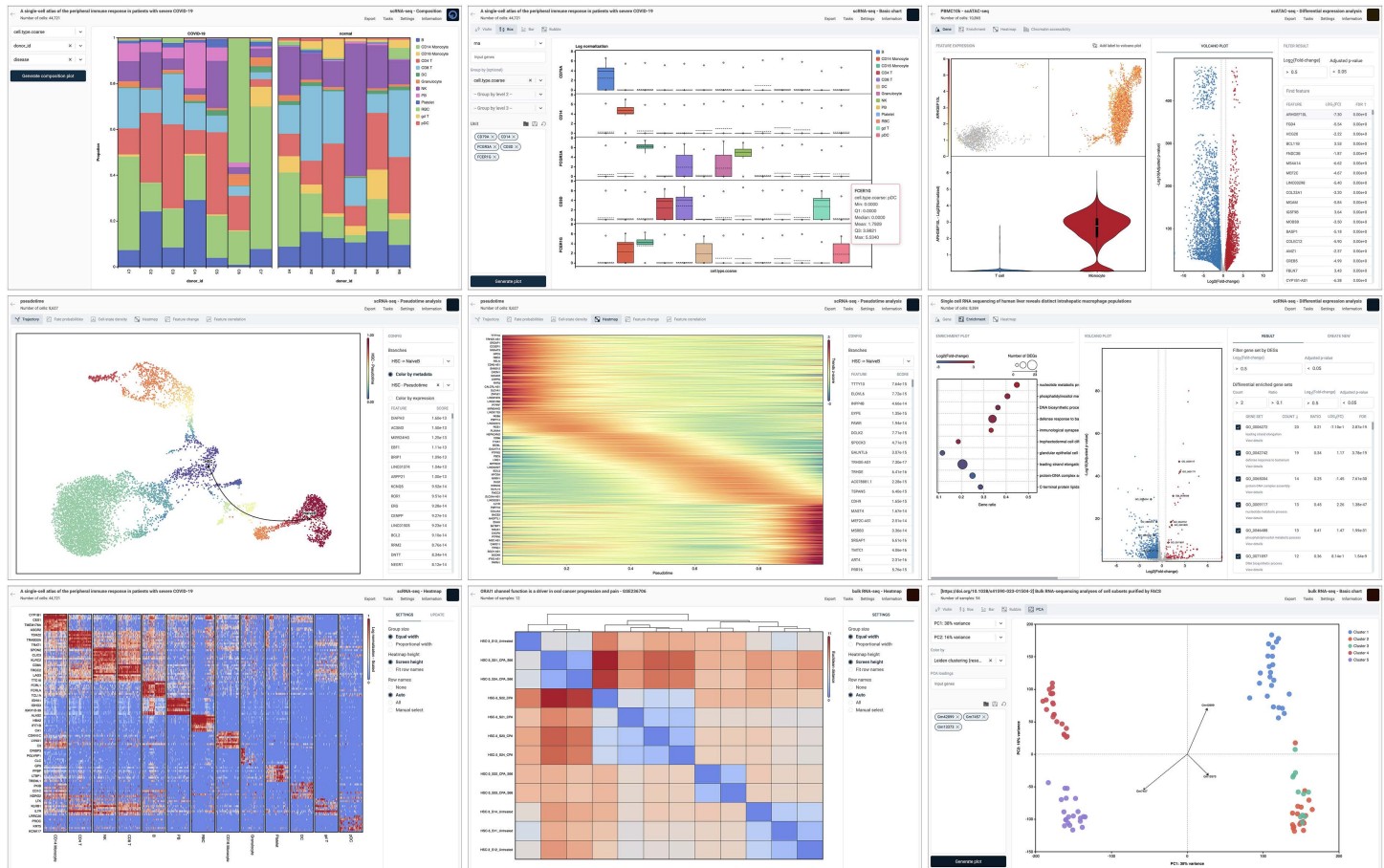

**Fig 3. OmnibusX presents analysis results through interactive plots, tables, and dashboards.** Shown here are representative dashboards from key analysis modules (composition, basic plots, DEG, trajectory, pseudotime heatmap, enrichment, marker heatmap, pairwise sample distance heatmap, PCA explanation, etc.) Each module includes exportable and customizable components.

platform supports comprehensive TCR/BCR repertoire analysis (Fig 4), including clonotype diversity metrics, repertoire overlap calculations via Jaccard similarity, and spectratype visualizations, facilitating insight into immune dynamics within single-cell datasets. For spatial transcriptomics, OmnibusX provides a cell–cell interaction analysis module that detects co-expression of ligand–receptor pairs curated by CellPhoneDB [12]. The interactions are further annotated with associated biological processes using Gene Ontology [13] (GO), facilitating interpretation of spatial communication within the tissue microenvironment.

## Plot customization for publication

A significant innovation of OmnibusX is its integrated plot-editing module, which allows full customization of nearly all generated visualizations directly within the application (Fig 4). This functionality enables researchers to produce publication-quality figures without relying on external software, significantly reducing the time required to move from analysis to manuscript preparation or presentation. Customization options include plot element configuration, axis control, font adjustments, color palette selection, legend formatting, element ordering, and layout refinement, allowing precise visual communication of analytical results.

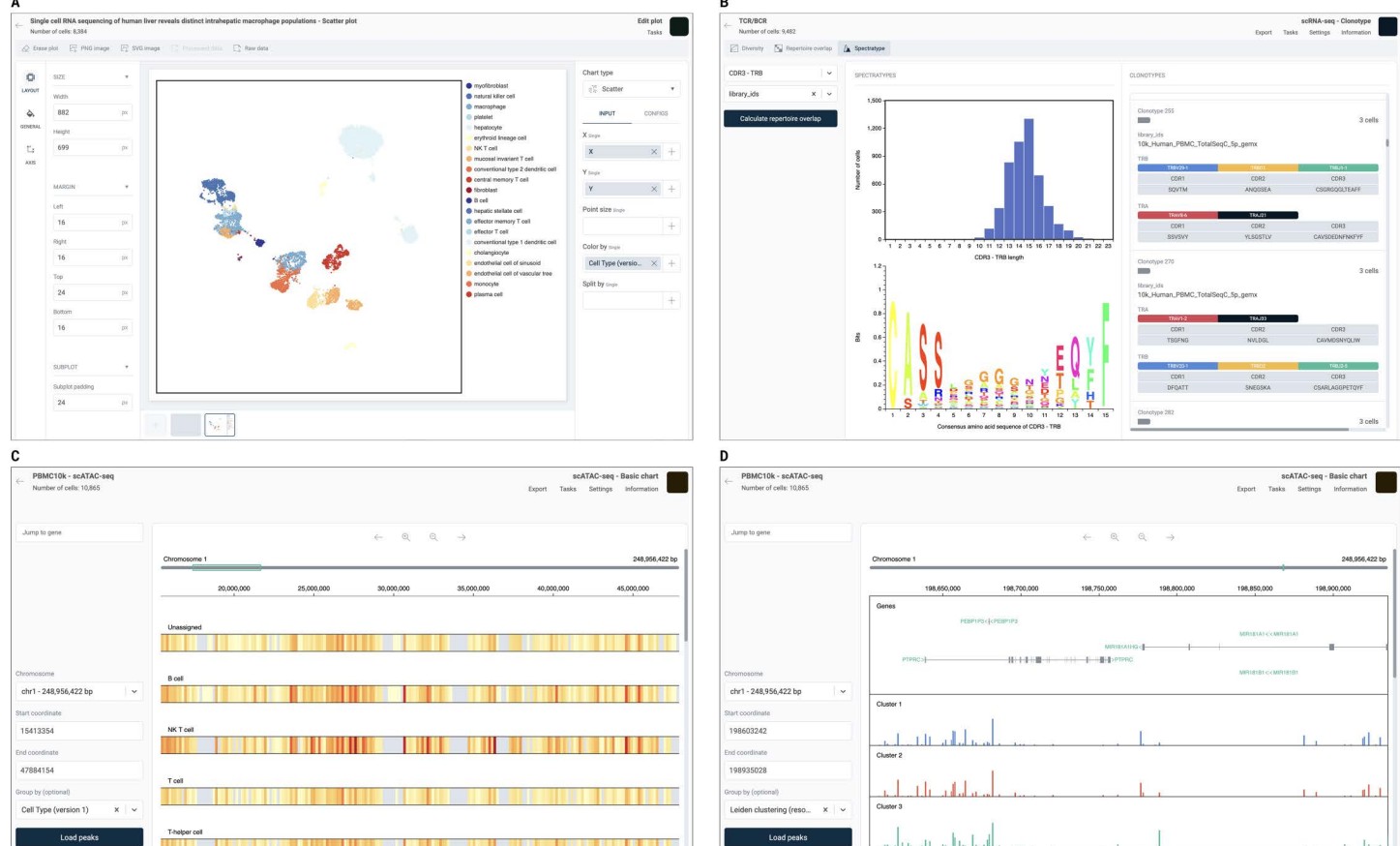

**Fig 4. OmnibusX provides specialized built-in modules to support advanced data interpretation and visualization. (A)** The plot editing module enables customization of nearly all plots generated during analysis. The right panel allows users to modify input variables and computation methods, while the left panel provides styling controls for layout, fonts, colors, and plot elements. **(B)** Clonotype analysis dashboard for TCR/BCR profiling. **(C)** Genome browser for scATAC-seq data in general mode: Chromatin accessibility is visualized as a heatmap across genomic regions, grouped by cell group labels, highlighting chromosome activity hotspots. **(D)** Genome browser in detailed view: Gene models (including introns, exons, and strand orientation) are displayed alongside cell group-specific chromatin accessibility tracks, shown as histograms. This mode facilitates high-resolution inspection of regulatory regions and gene-context relationships.

## Reproducibility and workflow management

OmnibusX automatically records all analysis parameters, visualization settings, and workflow states throughout the session. This ensures reproducibility, allowing researchers to revisit previous analyses, modify settings, or generate consistent figures without re-running computational steps. This built-in tracking enhances transparency, saves computational resources, and supports rigorous documentation of the analytical process.

## Group collaboration and data management

Collaborative research requires secure and structured data-sharing mechanisms. The enterprise edition of OmnibusX supports centralized data management with hierarchical user roles and group-based collaboration. Fine-grained permission settings (READ, WRITE, DELETE) enable administrators to define user access to shared datasets, enabling coordinated contributions from team members with different roles while maintaining data integrity and security. In addition to

group workspaces, each user is provided with a personal workspace where they retain full control over their submitted data. This structure ensures flexibility in project organization, supporting both private analysis and collaborative workflows.

This design accelerates collaboration among bioinformaticians, biologists, and pathologists. Advanced analyses can be executed externally by bioinformaticians, and the results can then be imported into OmnibusX for further exploration. Experimental scientists and clinical collaborators can interact directly with the processed data, annotate key findings, and extract relevant insights based on their specific research questions.

## Availability and future directions

OmnibusX is available as both a standalone desktop application and an enterprise edition for centralized deployment. A fully documented trial, along with licensing information, is accessible at https://omnibusx.com. All licensed users receive comprehensive documentation, responsive technical support, and optional personalized training to facilitate adoption and effective use. OmnibusX integrates core analytical functions from widely used open-source frameworks such as Seurat and Scanpy, ensuring performance consistent with these established tools, while proprietary modules—such as the cell type prediction engine—have been benchmarked in detail in our separate publication (Truong et al., 2024 [3]).

Future development will focus on expanding supported input formats, enhancing interoperability with emerging multi-omics technologies, and integrating advanced analytical modules, including cross-modal feature alignment, multi-view clustering, and multi-omics factor analysis, to enable deeper integration of scRNA-seq, scATAC-seq, proteomics, and spatial transcriptomics datasets. We also plan to incorporate additional spatial transcriptomics–specific methods, such as spatially variable gene detection, neighborhood enrichment analysis, and spatial autocorrelation metrics. Continued refinement of visualization capabilities, scalability for large datasets, and collaboration features will further strengthen OmnibusX as a unified, privacy-centric platform for multi-omics data analysis.

## Supporting information

**S1 Data. List of software packages and dependencies used in the OmnibusX platform.** The table includes package/library names and corresponding repository URLs.
(DOCX)

## Acknowledgments

We thank the OmnibusX user community, early adopters, institutional partners, and colleagues for invaluable feedback, collaboration, and ongoing support.

## Author contributions

Conceptualization, Methodology, Data curation, Formal analysis, Software, Writing – original draft, Writing – review & editing: Linh Truong, Thao Truong, Huy Nguyen. These authors contributed equally to this work.

## Author contributions

**Conceptualization:** Linh Truong, Thao Truong, Huy Nguyen.

**Data curation:** Linh Truong, Thao Truong, Huy Nguyen.

**Formal analysis:** Linh Truong, Thao Truong, Huy Nguyen.

**Methodology:** Linh Truong, Thao Truong, Huy Nguyen.

**Software:** Linh Truong, Thao Truong, Huy Nguyen.

**Writing – original draft:** Linh Truong, Thao Truong, Huy Nguyen.

**Writing – review & editing:** Linh Truong, Thao Truong, Huy Nguyen.

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
