## [Decision Letter · Decision Letter 0]

8 Aug 2025

PCOMPBIOL-D-25-01016

OmnibusX: A unified platform for accessible multi-omics analysis

PLOS Computational Biology

Dear Dr. Nguyen,

Thank you for submitting your manuscript to PLOS Computational Biology. After careful consideration, we feel that it has merit but does not fully meet PLOS Computational Biology's publication criteria as it currently stands. Therefore, we invite you to submit a revised version of the manuscript that addresses the points raised during the review process.

Please submit your revised manuscript within 60 days Oct 08 2025 11:59PM. If you will need more time than this to complete your revisions, please reply to this message or contact the journal office at ploscompbiol@plos.org. Please include the following items when submitting your revised manuscript:

We look forward to receiving your revised manuscript.

Kind regards,

Lun Hu

Academic Editor

PLOS Computational Biology

Ilya Ioshikhes

Section Editor

PLOS Computational Biology

**Additional Editor Comments:**

Although reviewers see the value of this work, they also had several major concerns that need to be resolved before publications. Particularly, authors should revise their manuscript by clarifying the availability of offline version, the extension of validation, and discussion of future work.

**Journal Requirements:**

1) Please provide an Author Summary. This should appear in your manuscript between the Abstract (if applicable) and the Introduction, and should be 150-200 words long. The aim should be to make your findings accessible to a wide audience that includes both scientists and non-scientists. Sample summaries can be found on our website under Submission Guidelines:

2) Your manuscript is missing the following sections: Abstract, and Availability and Future Directions. Please ensure that your article adheres to the standard Software article layout and order of Abstract, Introduction, Design and Implementation, Results, and Availability and Future Directions. For details on what each section should contain, see our Software article guidelines:

https://journals.plos.org/ploscompbiol/s/submission-guidelines#loc-software-submissions

**Reviewers' comments:**

Reviewer's Responses to Questions

**Comments to the Authors:**

Reviewer #1: This is an interesting paper that introduces a useful integrative tool for multi-omics data analysis. The platform could be very helpful for researchers, especially those who don’t have a background in programming. The focus on local data processing and privacy is also a strong point, especially for clinical or sensitive data.

However, there are a few inconsistencies that should be addressed:

1. The paper stresses the importance of avoiding cloud-based tools for privacy reasons, yet it mentions that OmnibusX connects to an AWS server for user authentication and downloading reference files. Doesn’t this AWS connection breach the very data security concerns the platform claims to avoid? Clarifying whether a fully offline mode is available would help.

2. The authors highlight transparency, but the platform includes proprietary tools like the cell type prediction engine. It’s unclear whether users have access to the code or functionality behind these tools. More explanation would be helpful.

3. The format support is a bit unclear. Seurat files are listed as supported, but later Seurat v5 is said to be unsupported. A clear list of supported formats would improve clarity.

4. Batch correction is described as being applied only for visualization (e.g., UMAP). It would be helpful to explain how this enhances UMAP plots by reducing batch effects and making cell populations more comparable across samples.

5. The provided link to the trial version (https://omnibusx.com/apps) was not working at the time of review. This should be checked and updated.

Fixing these issues would strengthen the manuscript and improve its usability.

Reviewer #2: Thank you for the opportunity to review your manuscript, “OmnibusX: A Unified Platform for Accessible Multi-Omics Analysis.” The platform you present addresses a clear and timely need in the life sciences by offering an integrated, privacy-conscious, and user-friendly environment for multi-omics analysis.

Strengths

- The wide support for diverse omics data types, including bulk and single-cell RNA-seq, scATAC-seq, spatial transcriptomics, TCR/BCR sequencing, and ADT, is impressive and highly valuable.

- The local, code-free architecture strikes a thoughtful balance between usability and data privacy, lowering barriers for non-programmers while supporting institutional compliance.

- Community-driven features and open access to curated resources enrich the user experience and foster collaborative development.

- The team’s commitment to continuous improvement and user feedback is commendable.

Suggestions for Improvement

- Benchmarking: Including performance benchmarks and comparisons with widely used tools (e.g., Seurat, Scanpy, Galaxy) within the manuscript would help contextualize OmnibusX’s strengths.

- Transparency: More information on proprietary modules (especially the cell-type annotation engine) would improve clarity.

References to validation studies or technical documentation would be helpful.

- User Impact: If available, data on adoption, user feedback, or usability assessments could further demonstrate accessibility and impact.

- Future Plans: A brief roadmap for future developments and how user feedback informs updates would be a valuable addition.

Reviewer #3: OmnibusX aims to overcome challenges posed by fragmented analytical tools and high computational barriers by integrating workflows for various technologies (including bulk RNA-seq, single-cell RNA-seq, single-cell ATAC-seq, and spatial transcriptomics) into a single, cohesive application. The application incorporates existing open-source tools (such as Scanpy, DESeq2, SciPy, and scikit-learn) into transparent, reproducible pipelines, providing users with control over analysis parameters. However, this article suffers from the following problems.

1. The paper mentions a proprietary cell type prediction module but fails to provide detailed information about its algorithmic principles, training data sources, and performance evaluation. Simply stating that it is "described in detail in a separate publication" is insufficient. The authors should at least outline the basic principles of the prediction engine, the type of model used, feature selection methods, and accuracy assessment results across different tissues and species.

2. Although the paper mentions support for spatial transcriptomics data analysis, the description of relevant methods is relatively simplistic and lacks in-depth discussion of spatial-specific analytical techniques. It is recommended that the authors expand their description of spatial statistical methods.

3. Despite the paper's title emphasizing "multi-omics," the methods section lacks specific technical descriptions of multimodal data integration analysis. It is recommended that the authors provide detailed explanations of how OmnibusX implements joint analysis of scRNA-seq with scATAC-seq, proteomics, or spatial transcriptomics data, particularly the implementation details of core technologies such as cross-modal feature alignment, multi-view clustering, and multi-omics factor analysis.

4. The paper acknowledges that complex analyses still rely on external bioinformatics experts, but fails to provide clear solutions for situations with limited local computational resources. It is recommended that the authors discuss how to maintain data privacy while clearly specifying recommended hardware configurations and performance expectations for various types of analyses, helping users assess whether their equipment is suitable for specific analytical tasks.

5. To further enrich the content of the paper and demonstrate awareness of research frontiers, it is suggested that the authors include more references to graph representation learning and the integration of bioinformatics methods with large models. For example: "Interpretable identification of cancer genes across biological networks via transformer-powered graph representation learning" and "LLM-DDI: Leveraging Large Language Models for Drug-Drug Interaction Prediction on Biomedical Knowledge Graph."

**Have the authors made all data and (if applicable) computational code underlying the findings in their manuscript fully available?**

Reviewer #1: **No: **

Reviewer #2: Yes

Reviewer #3: None

PLOS authors have the option to publish the peer review history of their article (what does this mean? ). If published, this will include your full peer review and any attached files.

**Do you want your identity to be public for this peer review?** For information about this choice, including consent withdrawal, please see our Privacy Policy .

Reviewer #1: No

Reviewer #2: **Yes: ** Nika Abdollahi

Reviewer #3: No

**Figure resubmission:**
---

## [Decision Letter · Decision Letter 1]

1 Sep 2025

Dear Mr Nguyen,

We are pleased to inform you that your manuscript 'OmnibusX: A unified platform for accessible multi-omics analysis' has been provisionally accepted for publication in PLOS Computational Biology.

Best regards,

Lun Hu

Academic Editor

PLOS Computational Biology

Ilya Ioshikhes

Section Editor

PLOS Computational Biology

Reviewer #1:

Reviewer #2:

Reviewer #3:

Reviewer's Responses to Questions

**Comments to the Authors:**

Reviewer #1: The authors have addressed all the concerns raised and the manuscript is of good quality for publication.

Reviewer #2: Dear Authors,

Thank you for your detailed responses to my comments. I am satisfied with the clarifications you provided and have no additional remarks.

Best regards,

Nika Abdollahi

Reviewer #3: The authors have solved the problem well.

**Have the authors made all data and (if applicable) computational code underlying the findings in their manuscript fully available?**

Reviewer #1: Yes

Reviewer #2: Yes

Reviewer #3: None

PLOS authors have the option to publish the peer review history of their article (what does this mean? ). If published, this will include your full peer review and any attached files.

**Do you want your identity to be public for this peer review?** For information about this choice, including consent withdrawal, please see our Privacy Policy .

Reviewer #1: **Yes: ** Namrata Bhattacharya

Reviewer #2: **Yes: ** Nika Abdollahi

Reviewer #3: No

---

## [Editor Report · Acceptance letter]

PCOMPBIOL-D-25-01016R1

OmnibusX: A unified platform for accessible multi-omics analysis

Dear Dr Nguyen,

I am pleased to inform you that your manuscript has been formally accepted for publication in PLOS Computational Biology. Your manuscript is now with our production department and you will be notified of the publication date in due course.

With kind regards,

Judit Kozma
